# Deceive D: Adaptive Pseudo Augmentation for GAN Training with Limited Data

**Liming Jiang**[1]    **Bo Dai**[1]    **Wayne Wu**[2]    **Chen Change Loy**[1]*

[1]S-Lab, Nanyang Technological University    [2]SenseTime Research

{liming002, bo.dai, ccloy}@ntu.edu.sg    wuwenyan@sensetime.com

## Abstract

Generative adversarial networks (GANs) typically require ample data for training in order to synthesize high-fidelity images. Recent studies have shown that training GANs with limited data remains formidable due to discriminator overfitting, the underlying cause that impedes the generator's convergence. This paper introduces a novel strategy called Adaptive Pseudo Augmentation (APA) to encourage healthy competition between the generator and the discriminator. As an alternative method to existing approaches that rely on standard data augmentations or model regularization, APA alleviates overfitting by employing the generator itself to augment the real data distribution with generated images, which deceives the discriminator adaptively. Extensive experiments demonstrate the effectiveness of APA in improving synthesis quality in the low-data regime. We provide a theoretical analysis to examine the convergence and rationality of our new training strategy. APA is simple and effective. It can be added seamlessly to powerful contemporary GANs, such as StyleGAN2, with negligible computational cost. Code: https://github.com/EndlessSora/DeceiveD.

## 1 Introduction

While state-of-the-art GANs like StyleGAN2 [20] are constantly pushing forward the fidelity and resolution of synthesized images, they usually require a large amount of training data to fully unleash their power. Training GANs with insufficient data tend to generate poor-quality images, as shown in Figure 1. Collecting sufficient data samples for these GANs is sometimes infeasible, especially in domains where data are sparse and privacy-sensitive. To ease the practical deployment of powerful GANs, it is necessary to devise new strategies for training GANs with limited data while preserving the quality of synthesis.

Recent studies have shown that the overfitting of the discriminator is the critical reason that impedes effective GAN training on limited data [2, 18, 50, 42], rendering severe instability of training dynamics. Specifically, when the discriminator starts to overfit, the distributions of its outputs for real and generated samples gradually diverge from each other [18, 42], and its feedback to the generator becomes less informative. Consequently, the generator converges to an inferior point, compromising the quality of synthesized images. Recent solutions to this problem include the use of standard data augmentations, either conventional or differentiable, to real and generated images [18, 50, 52, 41] or applying an additional model regularization term [42]. Addressing the discriminator overfitting is still an open problem. We are interested in finding an alternative way to the aforementioned approaches.

In this paper, we present a simple yet effective way to regularize a discriminator without introducing any external augmentations or regularization terms. We call our method *Adaptive Pseudo Augmentation* (APA). In contrast to previous standard data augmentations [18, 50, 52, 41], we exploit the

---

*Corresponding author.

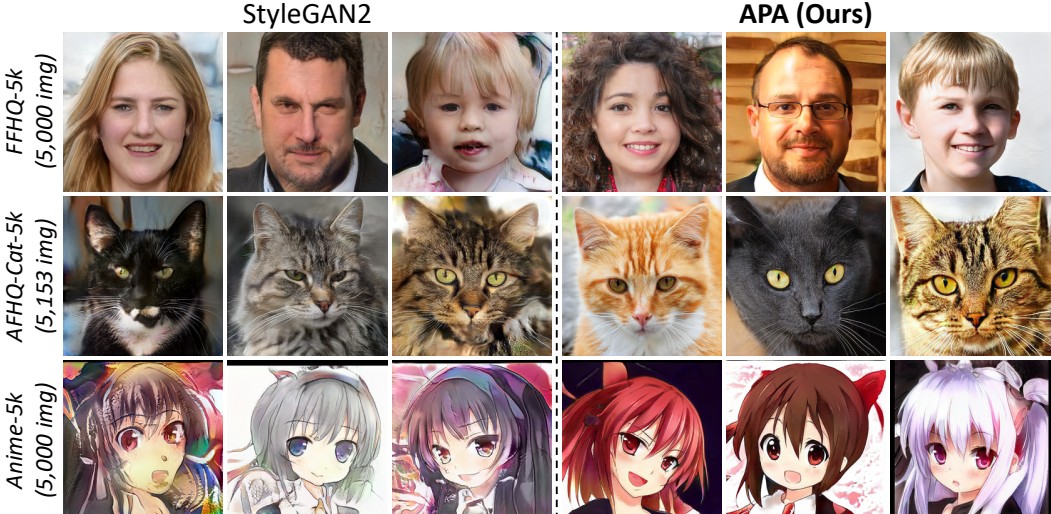

Figure 1: **StyleGAN2** [20] synthesized results (no truncation) deteriorate **given the limited amount of training data** ($256 \times 256$), *i.e.*, FFHQ [19] (a subset of $5,000$ images, $\sim 7\%$ of full data), AFHQ-Cat [8] ($5,153$ images, which is small by itself), and Danbooru2019 Portraits (Anime) [4] (a subset of $5,000$ images, $\sim 2\%$ of full data). The proposed Adaptive Pseudo Augmentation (APA) effectively ameliorates the degraded performance of StyleGAN2 on limited data.

generator in a GAN itself to provide the augmentation, a more natural way to regularize the overfitting of the discriminator. Compared to the model regularization, our approach is more adaptive to fit different settings and training status without manual tuning. Specifically, APA takes the fake/pseudo samples synthesized by the generator and moderately feeds them into the limited real data. Such pseudo data are adaptively presented to the discriminator as "real" instances. The goal of this pseudo augmentation for the real data is not to enlarge the real dataset but to suppress the discriminator's confidence in distinguishing real and fake distributions. The deceit is introduced adaptively, which is moderated by a deception probability according to the degree of overfitting. To quantify overfitting, we study a series of plausible heuristics derived from the discriminator raw output logits.

The main **contribution** of this work is a novel adaptive pseudo augmentation method for training GANs with limited data. This approach deceives the discriminator adaptively and mitigates the problem of discriminator overfitting. The proposed APA can be readily added to existing GAN training with negligible computational cost. We conduct extensive experiments to demonstrate the effectiveness of APA for state-of-the-art GAN training with limited data. The results are comparable or even better than other types of solutions [18, 42]. APA is also complementary to existing methods based on standard data augmentations for gaining a further performance boost. Besides, we theoretically connect APA with minimizing the JS divergence [23] between the smoothed data distribution and generated distribution, proving its convergence and rationality. We hope that our approach could extend the breadth and potential of solutions to GAN training with limited data.

## 2 Related Work

**Generative adversarial networks.** Generative adversarial networks (GANs) [9, 28, 35, 47] adopt an adversarial training scheme, where a generator keeps refining its capability in synthesizing images to compete with a discriminator (*i.e.*, a binary classifier) until the discriminator fails to classify the generated samples as fakes. GANs are known to suffer from training instability [9, 38, 2, 27]. Various approaches have been proposed to stabilize the training and improve the quality of synthesis by minimizing different $f$-divergences of the real and fake distributions [30]. The saturated form of vanilla GAN [9] is theoretically proven to minimize the JS divergence [23] between the two distributions. LSGAN [26] and EBGAN [49] correspond to the optimizations of $\chi^2$-divergence [34] and the total variation [3], respectively. On another note, WGAN [3] is designed for minimizing the Wasserstein distance.

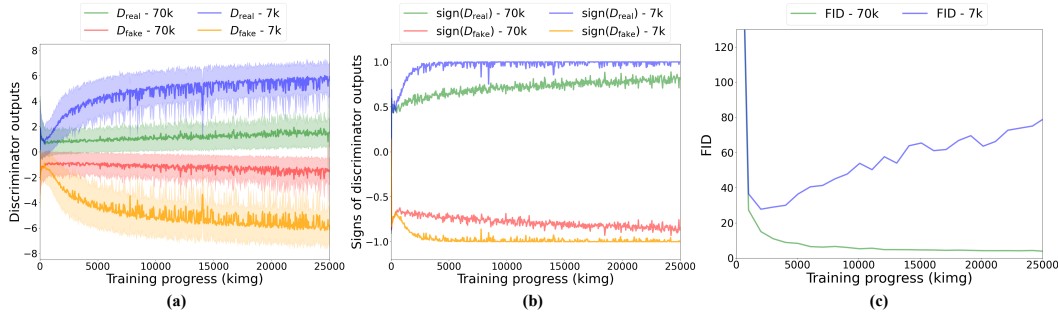

Figure 2: The overfitting of discriminator in GANs when limited training data are available. The three subplots report statistics of training snapshots of two StyleGAN2 [20] models on FFHQ [19] ($256 \times 256$). "70k" indicates the full dataset, and "7k" means a subset of $7,000$ images ($10\%$ data). The "kimg" denotes thousands of real images shown to the discriminator. (a) Discriminator raw output logits. (b) Signs of discriminator outputs. (c) Training convergence measured by FID [11].

State-of-the-art methods, such as PGGAN [17], BigGAN [5], StyleGAN [19], and StyleGAN2 [20], employ large-scale training with contemporary techniques, achieving photorealistic results. These methods have been extended to various tasks, including face generation [17, 19, 20], image editing [1, 7, 33], semantic image synthesis [44, 32, 25], image-to-image translation [13, 53, 8, 16, 15, 36], style transfer [24, 12, 22], and GAN inversion [31, 36, 46]. Despite the remarkable success, the performance of GANs relies heavily on the amount of training data.

**Training GANs with limited data.** The significance and difficulty of training GANs with limited data have been attracting attention from many researchers recently. The issue of data insufficiency tends to cause overfitting in the discriminator [45, 18, 50], which in turn deteriorates the stability of training dynamics in GANs, compromising the quality of generated images.

Many recent studies [48, 51, 41, 52, 50, 18] propose to apply standard data augmentations for GAN training to enrich the diversity of the dataset to mitigate the overfitting of the discriminator. For instance, DiffAugment (DA) [50] adopts the same differentiable augmentation to both real and fake images for the generator and the discriminator without manipulating the target distribution. Adaptive discriminator augmentation (ADA) [18] shares a similar idea with DA, while it further devises an adaptive approach that controls the strength of data augmentations adaptively. In this work, we extend the study of such an adaptive approach.

Another type of solution is model regularization. Previous efforts on regularizing GANs include adding noise to the inputs of the discriminator [2, 40, 14], gradient penalty [27, 10, 37], one-sided label smoothing [38], spectral normalization [29], label noise [6], *etc*. These methods are designed for stabilizing training or preventing mode collapse [38]. The essence of their goals could be considered similar to our method since training GANs in the low-data regime exhibits similar behaviors as previously observed in early GANs with sufficient data. Additional discussions on these techniques are provided in the *supplementary material*. Under the limited data setting, a very recent study proposes an LC-regularization term [42] to regulate the discriminator predictions using two exponential moving average variables that track the discriminator outputs throughout training.

Our work explores an alternative solution from a different perspective, which is also complementary to previous approaches based on standard data augmentations.

## 3 Methodology

In GAN's adversarial training, the goal of the generator $G$ is to deceive the discriminator $D$ and maximize the probability that $D$ makes a wrong judgment. Therefore, $G$ keeps refining its generated samples to better deceive $D$ over time. When the training only accesses a limited amount of data, one would observe that $D$ turns out to be overly confident and hardly makes any mistake, causing its feedback to $G$ to become meaningless.

In Figure 2, we show the training "snapshots" of two StyleGAN2 [20] models on the FFHQ dataset [19]. The settings of the two models differ only by the amount of data available to them for

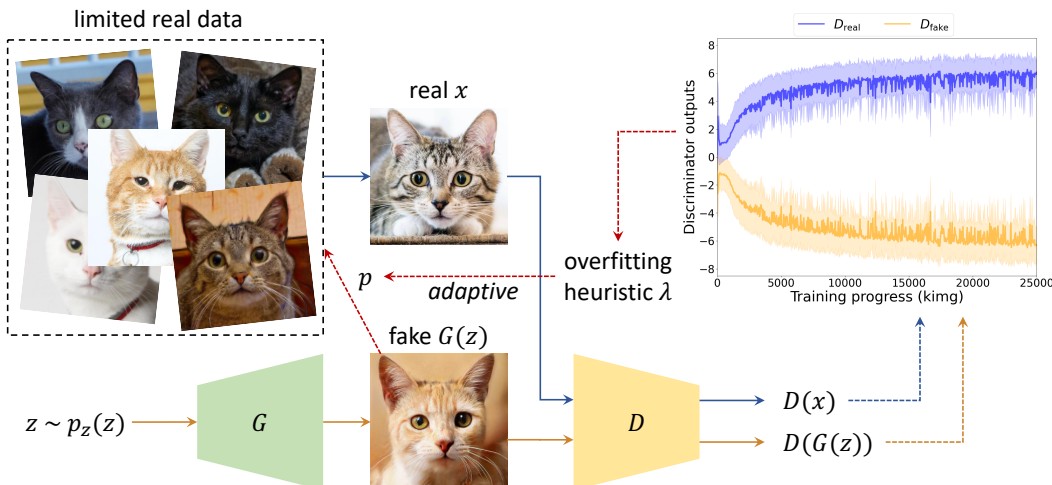

Figure 3: Adaptive pseudo augmentation (APA) for GAN training with limited data. We employ a GAN to augment itself using the generated images to deceive the discriminator adaptively. Specifically, APA feeds the images synthesized by the generator into the limited real data moderately, and these fakes are presented as "real" instances to the discriminator. Such deceits are introduced adaptively using an overfitting heuristic $\lambda$ defined by the discriminator raw output logits. The augmentation/deception probability $p$ can be adaptively controlled throughout training.

training. As can be observed, both training processes start smoothly, and distributions of discriminator outputs for the real and generated images overlap at the early stage. As the training progresses, the discriminator, which only has access to limited data (7k images, $10\%$ of full data), experiences diverged predictions much more rapidly, and the average sign boundary turns out to be more apparent. This divergence in prediction shows that $D$ becomes increasingly confident in classifying real and fake samples. At the late stage of training, $D$ can even judge all the input samples correctly with high confidence. Meanwhile, the evaluation FID [11] scores (lower is better) deteriorate, consistent with the divergence of $D$'s predictions. The phenomena above demonstrate how a discriminator gets overfitted quickly with limited data. As can be seen from the FID curves, the overfitting directly influences training dynamics and the convergence of $G$, leading to poor generation performance.

## 3.1 Adaptive Pseudo Augmentation

The generator itself naturally possesses the capability to counteract discriminator overfitting. To harness this capability, our method employs a GAN to augment itself using the generated samples (see Figure 3). Specifically, we feed the samples generated by $G$ into the limited real data to form a pseudo-real set. The fake images in this set will be adaptively presented to $D$, pretending themselves as real data. The goal here is to deceive $D$ with the pseudo-real set and consequently suppressing its confidence in distinguishing real and fake images.

Blindly presenting the fake images as reals to $D$ may weaken the fundamental ability of $D$ in adversarial training. In our approach, the deceit is introduced adaptively to avoid any potential adverse effects. To moderate the deception, we perform pseudo augmentation based on a probability, $p \in [0, 1)$, that quantifies the deception strength. Specifically, the pseudo augmentation will be applied with the probability $p$ or be skipped with the probability $1 - p$.

We note that the overfitting state of $D$ is dynamic throughout the training (see Figure 2). It is intuitive to let the deception probability $p$ be adjusted adaptively based on the degree of overfitting. Ideally, $p$ should be adjusted without manual tuning regardless of data scales and properties. To achieve this goal, inspired by ADA [18], we apply an overfitting heuristic $\lambda$ that quantifies the degree of $D$'s overfitting. We extend the control scheme of ADA and provide three plausible variants:

$$\lambda_r = \mathbb{E}\left(\text{sign}\left(D_{\text{real}}\right)\right), \quad \lambda_f = -\mathbb{E}\left(\text{sign}\left(D_{\text{fake}}\right)\right), \quad \lambda_{rf} = \frac{\mathbb{E}\left(\text{sign}\left(D_{\text{real}}\right)\right) - \mathbb{E}\left(\text{sign}\left(D_{\text{fake}}\right)\right)}{2}, \quad (1)$$

where $D_{\text{real}}$ and $D_{\text{fake}}$ are defined as

$$D_{\text{real}} = \text{logit}\left(D\left(x\right)\right), \quad D_{\text{fake}} = \text{logit}\left(D\left(G\left(z\right)\right)\right), \quad (2)$$

where $\mathrm{logit}$ denotes the logit function. As shown in Figure 2, the $\lambda_r$ in Eq. (1) estimates the portion of real images that obtain positive logit predictions by $D$, and that of generated images is captured by $\lambda_f$. Besides, $\lambda_{rf}$ indicates half of the distance between the signs of the real and fake logits. For all these heuristics, $\lambda = 0$ represents no overfitting, and $\lambda = 1$ means complete overfitting. We use $\lambda_r$ in our main experiments and study other variants in the ablation study.

The strategy of using $\lambda$ to adjust $p$ is as follows. We set a threshold $t$ (in most cases of our experiments, $t = 0.6$) and initialize $p$ to be zero. If $\lambda$ signifies too much/little overfitting regarding $t$ (*i.e.*, larger/smaller than $t$), the probability $p$ will be increased/decreased by one fixed step. Using this step size, $p$ can increase from zero to one in $500\mathrm{k}$ images shown to $D$. We adjust $p$ once every four iterations and clamp $p$ from below to zero after each adjustment. In this way, the strength of pseudo augmentation can be adaptively controlled based on the degree of overfitting (see Figure 3).

## 3.2 Theoretical Analysis

Let $p_z(z)$ be the prior on the input noise variable. The mapping from the latent space to the image space is denoted as $G(z)$. For sample $x$, $D(x)$ represents the estimated probability of $x$ coming from the real data. To examine the rationality of APA, we analyze it in a non-parametric setting [9], where a model is represented with infinite capacity by exploring its convergence in the space of probability density functions. Ideally, the estimated probability distribution $p_g$ defined by $G$ should perfectly model the real data distribution $p_{\mathrm{data}}$ without bias if given enough capability and training time.

Since the deception strength $p$ is adaptively adjusted, to facilitate this theoretical analysis, we make a mild assumption that $\alpha$ is the expected strength that approximates the effect of dynamic adjustment during the entire training procedure. Since $p \in [0, 1)$, we have $0 \leq \alpha < p_{\max} < 1$, where $p_{\max}$ is the maximum of deception strength throughout training. Hence, the value function $V(G, D)$ for the minimax two-player game of APA can be reformulated as:

$$\min_G \max_D V(G, D) = (1 - \alpha) \, \mathbb{E}_{x \sim p_{\mathrm{data}}(x)} \left[\log D(x)\right] + \alpha \, \mathbb{E}_{z \sim p_z(z)} \left[\log D(G(z))\right]$$
$$+ \mathbb{E}_{z \sim p_z(z)} \left[\log \left(1 - D(G(z))\right)\right], \tag{3}$$

First, let us consider the optimal discriminator [9] for any given generator.

**Proposition 1.** *If the generator $G$ is fixed, the optimal discriminator $D$ for APA is:*

$$D_G^*(x) = \frac{(1 - \alpha) \, p_{\mathrm{data}}(x) + \alpha \, p_g(x)}{(1 - \alpha) \, p_{\mathrm{data}}(x) + (1 + \alpha) \, p_g(x)} \tag{4}$$

*Proof.* Applying APA, given any generator $G$, the training objective of the discriminator $D$ is to maximize the value function $V(G, D)$ in Eq. (3):

$$V(G, D) = (1 - \alpha) \int_x p_{\mathrm{data}}(x) \log D(x) dx + \alpha \int_z p_z(z) \log D(g(z)) dz + \int_z p_z(z) \log \left(1 - D(g(z))\right) dz$$

$$= \int_x \left[(1 - \alpha) \, p_{\mathrm{data}}(x) \log D(x) + \alpha \, p_g(x) \log D(x) + p_g(x) \log \left(1 - D(x)\right)\right] dx$$

$$= \int_x \left[\left((1 - \alpha) \, p_{\mathrm{data}}(x) + \alpha \, p_g(x)\right) \log D(x) + p_g(x) \log \left(1 - D(x)\right)\right] dx \tag{5}$$

For any $(m, n) \in \mathbb{R}^2 \setminus \{0, 0\}$, the function $f(y) = m \log(y) + n \log(1 - y)$ achieves its maximum in the range $[0, 1]$ at $\frac{m}{m+n}$. Besides, the discriminator $D$ is defined only inside of $\mathrm{supp}(p_{\mathrm{data}}) \cup \mathrm{supp}(p_g)$, where $\mathrm{supp}$ is the set-theoretic support. Therefore, we conclude the proof for Proposition 1.

We have got the optimal discriminator $D_G^*(x)$ in Eq. (4) that maximizes the value function $V(G, D)$ given any fixed generator $G$. The goal of generator $G$ in the adversarial training is to minimize the value function $V(G, D)$ in Eq. (3) when $D$ achieves the optimum. Since the training objective of $D$ can be interpreted as maximizing the log-likelihood for the conditional probability $P(Y = y|x)$, where $Y$ estimates that $x$ comes from $p_{\mathrm{data}}$ (*i.e.*, $y = 1$) or from $p_g$ (*i.e.*, $y = 0$), we reformulate virtual training criterion [9] as:

$$C(G) = (1 - \alpha) \, \mathbb{E}_{x \sim p_{\mathrm{data}}} \left[\log D_G^*(x)\right] + \alpha \, \mathbb{E}_{z \sim p_z} \left[\log D_G^*(G(z))\right] + \mathbb{E}_{z \sim p_z} \left[\log \left(1 - D_G^*(G(z))\right)\right]$$
$$= (1 - \alpha) \, \mathbb{E}_{x \sim p_{\mathrm{data}}} \left[\log D_G^*(x)\right] + \alpha \, \mathbb{E}_{x \sim p_g} \left[\log D_G^*(x)\right] + \mathbb{E}_{x \sim p_g} \left[\log \left(1 - D_G^*(x)\right)\right] \tag{6}$$

Then, let us consider the global minimum of $C(G)$ trained with the proposed APA.

**Proposition 2.** *Applying APA, the global minimum of the virtual training criterion $C(G)$ is still achieved if and only if $p_g = p_{\text{data}}$, where $C(G) = -\log 4$.*

*Proof.* 1) If $p_g = p_{\text{data}}$, we have $D_G^*(x) = \frac{1}{2}$ according to Eq. (4). By inspecting Eq. (6) at $D_G^*(x) = \frac{1}{2}$, we get $C^*(G) = (1-\alpha)\log\frac{1}{2} + \alpha\log\frac{1}{2} + \log\frac{1}{2} = \log\frac{1}{2} + \log\frac{1}{2} = -\log 4$.

2) To verify $C^*(G)$ is the global minimum of $C(G)$, and it can only be achieved when $p_g = p_{\text{data}}$, as in the derivation of Eq. (5), we obtain:

$$C(G) = \int_x ((1-\alpha)\,p_{\text{data}}(x) + \alpha\,p_g(x))\log D_G^*(x)dx + \int_x p_g(x)\log(1 - D_G^*(x))dx \quad (7)$$

Observe that

$$-\log 4 = (1-\alpha)\,\mathbb{E}_{x\sim p_{\text{data}}}[-\log 2] + \alpha\,\mathbb{E}_{x\sim p_g}[-\log 2] + \mathbb{E}_{x\sim p_g}[-\log 2]$$
$$= -\int_x ((1-\alpha)\,p_{\text{data}}(x) + \alpha\,p_g(x))\log 2\,dx - \int_x p_g(x)\log 2\,dx \quad (8)$$

Subtracting Eq. (8) from Eq. (7),

$$C(G) = -\log 4 + \int_x ((1-\alpha)\,p_{\text{data}}(x) + \alpha\,p_g(x))\log 2\cdot D_G^*(x)dx + \int_x p_g(x)\log 2\cdot(1 - D_G^*(x))dx \quad (9)$$

By substituting Eq. (4) into Eq. (9), we achieve:

$$C(G) = -\log 4 + \text{KLD}\left(((1-\alpha)\,p_{\text{data}} + \alpha\,p_g)\,\middle\|\,\frac{(1-\alpha)\,p_{\text{data}} + (1+\alpha)\,p_g}{2}\right)$$
$$+ \text{KLD}\left(p_g\,\middle\|\,\frac{(1-\alpha)\,p_{\text{data}} + (1+\alpha)\,p_g}{2}\right), \quad (10)$$

where KLD is the Kullback-Leibler (KL) divergence. Moreover, Eq. (10) further implies that the generation process of $G$ by APA can be regarded as minimizing the Jensen-Shannon (JS) divergence between the smoothed data distribution and the generated distribution:

$$C(G) = -\log 4 + 2\cdot\text{JSD}\left(((1-\alpha)\,p_{\text{data}} + \alpha\,p_g)\,\|\,p_g\right). \quad (11)$$

For the two distributions $P$ and $Q$, their JS divergence $\text{JSD}(P\|Q) \geq 0$ and $\text{JSD}(P\|Q) = 0$ if and only if $P = Q$. Therefore, for $0 \leq \alpha < p_{\max} < 1$, we obtain that $C^*(G) = -\log 4$ is the global minimum of $C(G)$, and the only solution is $(1-\alpha)\,p_{\text{data}} + \alpha\,p_g = p_g$, *i.e.*, $p_g = p_{\text{data}}$. Q.E.D.

Given the proof in [9], if $G$ and $D$ have enough capacity to reach their optimum, Proposition 2 indicates that the generated distribution $p_g$ can ideally converge to the real data distribution $p_{\text{data}}$. So far, we have proved the convergence of $G$ trained with our proposed APA, which can perfectly model the real data distribution given sufficient capability and training time. Besides, the JS divergence term between the smoothed data distribution and the generated distribution in Eq. (11) implies that the judgment of $D$ may be moderated to alleviate overfitting. These conclusions explain the rationality of the proposed APA for training GANs with limited data.

## 4  Experiments

**Datasets.** We use four datasets in our main experiments: Flickr-Faces-HQ (FFHQ) [19] with $70,000$ human face images, AFHQ-Cat [8] with $5,153$ cat faces, Caltech-UCSD Birds-200-2011 (CUB) [43] with $11,788$ images of birds, and Danbooru2019 Portraits (Anime) [4] with $302,652$ anime portraits. We exploit some of their artificially limited subsets under different settings. All the images are resized to a moderate resolution of $256 \times 256$ using a high-quality Lanczos filter [21] to reduce the energy consumption for large-scale GAN training while preserving image quality. Additional dataset details, including the data source and license information, can be found in our *supplementary material*.

**Evaluation metrics.** We follow the standard evaluation protocol [5, 42] for the quantitative evaluation. Specifically, we use the Fréchet Inception Distance (FID, lower is better) [11], which quantifies the distance between distributions for the real and generated images. FID evaluates the realness of synthesized images. Following [11, 18], we calculate FID for the models trained with limited

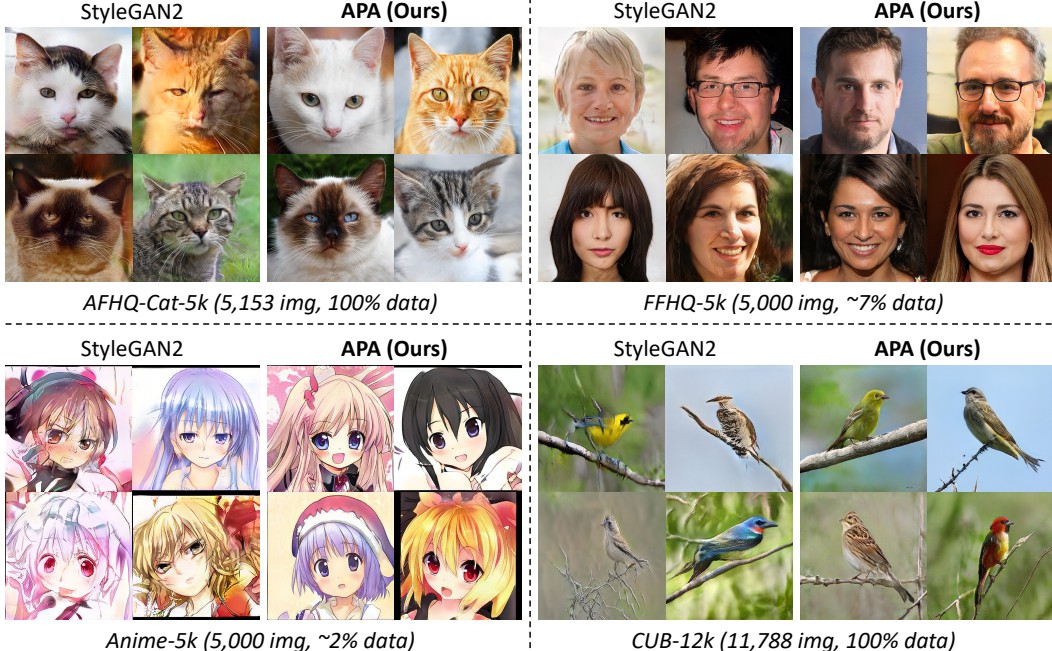

Figure 4: The proposed APA improves StyleGAN2 [20] synthesized results ($256 \times 256$, no truncation) on **various datasets** with limited data amounts. We randomly select subsets to confine the size of large datasets (*i.e.*, FFHQ-5k [19] and Anime-5k [4]) and directly use small datasets (*i.e.*, AFHQ-Cat-5k [8] and CUB-12k [43]) whose data amount is already insufficient for StyleGAN2.

Table 1: The FID (lower is better) and IS (higher is better) scores ($256 \times 256$) of our method compared to state-of-the-art StyleGAN2 on **various datasets** with limited data amounts.

| Method | AFHQ-Cat-5k | | FFHQ-5k | | Anime-5k | | CUB-12k | |
|---|---|---|---|---|---|---|---|---|
| | FID $\downarrow$ | IS $\uparrow$ | FID $\downarrow$ | IS $\uparrow$ | FID $\downarrow$ | IS $\uparrow$ | FID $\downarrow$ | IS $\uparrow$ |
| StyleGAN2 [20] | 7.737 | 1.825 | 37.830 | 4.018 | 23.778 | 2.289 | 23.437 | 5.812 |
| APA (Ours) | **4.876** | **2.156** | **13.249** | **4.487** | **13.089** | **2.330** | **12.889** | **5.869** |

datasets using 50k generated images and all the real images in the original datasets. We also apply the Inception Score (IS, higher is better) [38]. IS considers the clarity and diversity of generated images.

**Implementation details.** We choose the state-of-the-art StyleGAN2 [20] as the backbone to verify the effectiveness of APA on limited data. We use the default setups of APA provided in Section 3.1 unless specified otherwise. For a fair and controllable comparison, we reimplement all baselines and run the experiments from scratch using official code. All the models are trained on 8 NVIDIA Tesla V100 GPUs. Please refer to the *supplementary material* for more implementation details and additional benchmark results (*e.g.*, performance on BigGAN [5]).

### 4.1 The Effectiveness of APA

**Effectiveness on various datasets.** The comparative results of StyleGAN2 on various datasets with limited data amounts are shown in Figure 4. The quality of images synthesized by StyleGAN2 deteriorates under limited data. Ripple artifacts appear on the cat faces and human faces, and the facial features of the anime faces are misplaced. On the bird dataset with heavy background clutter, the generated images are completely distorted albeit trained with more data. The proposed APA significantly ameliorates image quality on all these datasets, producing much more photorealistic results. The quantitative evaluation results are reported in Table 1. Applying APA contributes to a performance boost of FID and IS in all cases, suggesting that the synthesized images are with higher quality and diversity on different datasets.

**Effectiveness given different data amounts.** The comparative results on subsets of FFHQ [19], with varying amounts of data, are shown in Figure 5. The corresponding quantitative test results are

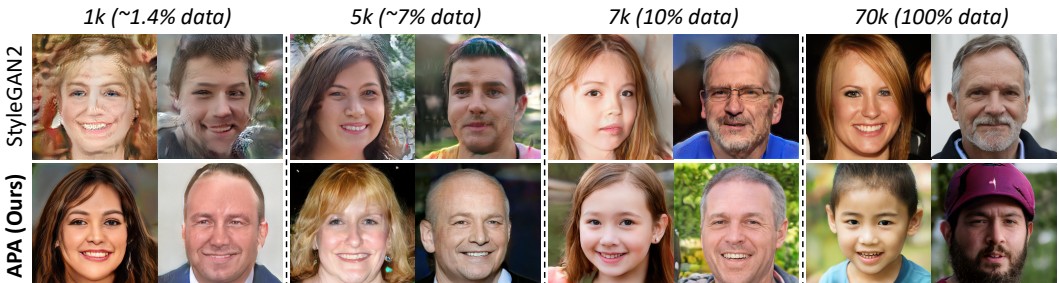

Figure 5: The effectiveness of APA to improve StyleGAN2 [20] synthesized results ($256 \times 256$, no truncation) on the subsets of FFHQ [19] with **different data amounts**.

Table 2: The FID (lower is better) and IS (higher is better) scores ($256 \times 256$) of our method on StyleGAN2 trained using the subsets of FFHQ [19] with **different data amounts**.

| Method | 1k ($\sim 1.4\%$) | | 5k ($\sim 7\%$) | | 7k (10%) | | 70k (100%) | |
| | FID $\downarrow$ | IS $\uparrow$ | FID $\downarrow$ | IS $\uparrow$ | FID $\downarrow$ | IS $\uparrow$ | FID $\downarrow$ | IS $\uparrow$ |
|---|---|---|---|---|---|---|---|---|
| StyleGAN2 [20] | 86.407 | 2.806 | 37.830 | 4.018 | 27.738 | 4.264 | 3.862 | 5.243 |
| APA (Ours) | **45.192** | **4.130** | **13.249** | **4.487** | **10.800** | **4.860** | **3.678** | **5.336** |

presented in Table 2. APA improves the image quality and metric performance in all cases. Notably, the quality of synthesized images by APA on 5k/7k data is visually close to StyleGAN2 results on the full dataset while with an order of magnitude fewer training samples. As for the quantitative results, the IS score of APA on 1k data is better than that of StyleGAN2 on 5k data, and both metrics of APA on 5k data outperform StyleGAN2 results on 7k data. APA can even improve StyleGAN2 performance on the full dataset, further indicating its potential.

**Overfitting and convergence analysis.** As shown in Figure 6, the divergence of StyleGAN2 discriminator predictions can be effectively restricted on FFHQ-7k (10% of full data) by applying the proposed APA. The curves of APA on FFHQ-7k become closer to that of StyleGAN2 on FFHQ-70k, suggesting the effectiveness of APA in curbing the overfitting of the discriminator. Besides, APA improves the training convergence of StyleGAN2 on limited data, shown by the FID curves. More overfitting and convergence analysis can be found in the *supplementary material*.

## 4.2 Comparison with Other Solutions for GAN Training with Limited Data

**Performance and compatibility.** We compare the proposed APA with representative approaches designed for the low-data regime, including ADA [18] and LC-regularization (LC-Reg) [42], which perform standard data augmentations and model regularization, respectively. The results are reported in Table 3. As a single method, APA outperforms previous solutions in most cases, effectively

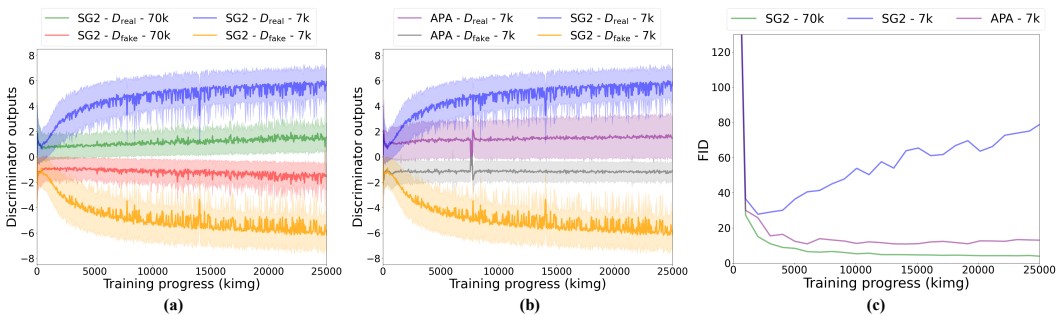

Figure 6: The **overfitting and convergence status** of APA compared to StyleGAN2 (SG2) on FFHQ [19] ($256 \times 256$). (a) The discriminator raw output logits of StyleGAN2 on the full (70k) or limited (7k) datasets. (b) The discriminator raw output logits of StyleGAN2 and APA on the limited (7k) dataset. (c) The training convergence shown by FID.

Table 3: The FID (lower is better) and IS (higher is better) scores ($256 \times 256$) of our method **compared to other state-of-the-art solutions designed for GAN training with limited data** on StyleGAN2. The bold number indicates the best value, and the underline marks the second best.

| Method | AFHQ-Cat-5k | | FFHQ-5k | | FFHQ-70k (full) | |
|---|---|---|---|---|---|---|
| | FID $\downarrow$ | IS $\uparrow$ | FID $\downarrow$ | IS $\uparrow$ | FID $\downarrow$ | IS $\uparrow$ |
| StyleGAN2 [20] | 7.737 | 1.825 | 37.830 | 4.018 | 3.862 | 5.243 |
| ADA [18] | 6.053 | 2.119 | 11.409 | 4.721 | 4.018 | 5.329 |
| LC-Reg [42] | 6.699 | 1.943 | 35.148 | 3.926 | 3.933 | 5.312 |
| APA (Ours) | 4.876 | 2.156 | 13.249 | 4.487 | **3.678** | **5.336** |
| ADA + APA (Ours) | **4.377** | **2.169** | **8.379** | **4.849** | 3.811 | 5.321 |

Table 4: The FID (lower is better) and IS (higher is better) scores ($256 \times 256$) of our method **compared to previous techniques for regularizing GANs** on StyleGAN2 trained with FFHQ-5k [19].

| Metric | StyleGAN2 [20] | Instance noise [40] | One-sided LS [38] | APA (Ours) |
|---|---|---|---|---|
| FID $\downarrow$ | 37.830 | 40.981 | 33.978 | **13.249** |
| IS $\uparrow$ | 4.018 | 4.231 | 4.029 | **4.487** |

improving the StyleGAN2 baseline on both the limited and full datasets. Although ADA [18] achieves slightly better results than our method on FFHQ-5k, it yields a worse FID score with StyleGAN2 on the full dataset. Applying LC-Reg [42] needs careful manual tuning, and its own effect on limited data is not apparent compared to other methods.

It is noteworthy that APA is also complementary to existing methods based on standard data augmentations, *e.g.*, ADA [18]. As can be observed in Table 3, APA can further boost the performance of StyleGAN2 with ADA [18] given limited training data, suggesting the compatibility of our approach with standard data augmentations. Combining ADA [18] and APA on FFHQ full data outperforms StyleGAN2 but is slightly inferior to applying APA solely. The degraded performance is mainly affected by ADA [18], which we empirically found might slightly harm the performance when the training data is sufficient. Overall, these methods are not dedicated to improving performance under sufficient data. Nevertheless, as a beneficial side effect, the proposed APA may have this potential. More comparative results are included in our *supplementary material*.

**Training cost.** We compare the computational cost of APA against ADA [18], using the same basic official codebase. There is no parameter or memory increment for both methods. As for the time consumption, we test the training cost on 8 NVIDIA Tesla V100 GPUs. On FFHQ-5k ($256 \times 256$), the average training time of the StyleGAN2 [20] baseline is $(4.740 \pm 0.100)$ sec/kimg (*i.e.*, seconds per thousand of images shown to the discriminator). The cost of our method is negligible, slightly increasing this value to $(4.789 \pm 0.078)$ sec/kimg. As a reference, the value for ADA [18] is $(5.327 \pm 0.116)$ sec/kimg, spending additional time for applying external augmentations.

### 4.3 Comparison with Previous Techniques for Regularizing GANs

As mentioned in Section 2, APA is closely related to previous techniques for regularizing GANs. The comparative results on APA and some representative conventional techniques, *i.e.*, instance noise [40] and one-sided label smoothing (LS) [38], are shown in Table 4. Applying instance noise [40] may not boost the performance of StyleGAN2 much under limited data. One-sided label smoothing (LS) [38] (with the real label of $0.9$) outperforms StyleGAN2 but still has a huge performance gap with our method. This further suggests the effectiveness and usefulness of APA.

### 4.4 Ablation Studies

**Ablation studies on variants of APA.** We study three key elements of APA, *i.e.*, the overfitting heuristic $\lambda$, the deception strength $p$, and the deception strategy. The version used in our main experiments is denoted as the "main" version, where $\lambda = \lambda_r$, and $p$ is adjusted adaptively. Besides, the "main" version applies the deception strategy that is analogous to one-sided label flipping. As reported in Table 5, when using other variants of $\lambda$ we suggested in Eq. (1) (*i.e.*, $\lambda = \lambda_f$ and $\lambda = \lambda_{rf}$), the models achieve comparable performance as the main version. The FID score becomes even better for $\lambda = \lambda_{rf}$, indicating the flexibility of our provided heuristics to extend and modify APA. More interestingly, even a fixed moderate deception probability (*e.g.*, $p = 0.5$) can still work much better

Table 5: **Ablation studies on variants of APA** on FFHQ-5k [19] ($256 \times 256$). We study three key elements of APA, *i.e.*, the overfitting heuristic $\lambda$, the deception strength $p$, and the deception strategy. The "main" denotes the main version used in our previous experiments (*i.e.*, $\lambda = \lambda_r$, $p$ is adaptively adjusted, and the deception strategy is analogous to one-sided label flipping).

| Metric | StyleGAN2 | main | $\lambda = \lambda_f$ | $\lambda = \lambda_{rf}$ | $p = 0.5$ (fix) | two-sided |
|--------|-----------|------|-----------------------|--------------------------|-----------------|-----------|
| FID $\downarrow$ | 37.830 | 13.249 | 13.470 | **12.679** | 14.632 | 15.440 |
| IS $\uparrow$ | 4.018 | **4.487** | 4.420 | 4.412 | 4.403 | 4.167 |

Table 6: **Ablation studies on the threshold $t$** on FFHQ-5k [19] ($256 \times 256$). The "main" denotes the main version used in our previous experiments (*i.e.*, $t = 0.6$).

| Metric | StyleGAN2 | $t = 0.4$ | $t = 0.6$ (main) | $t = 0.8$ |
|--------|-----------|-----------|------------------|-----------|
| FID $\downarrow$ | 37.830 | 13.687 | **13.249** | 13.984 |
| IS $\uparrow$ | 4.018 | 4.418 | **4.487** | 4.395 |

than original StyleGAN2 on limited data, albeit slightly inferior to the adaptively adjusted $p$. This implies the importance of the pseudo augmentation, and the adaptive control scheme can further boost performance without manual tuning. As for the deception strategy, we empirically observe that two-sided label flipping can still outperform StyleGAN2 but is inferior to the main version.

**Ablation studies on the threshold $t$.** We further provide the ablation studies on the threshold value $t$ in Table 6. The version used in our main experiments is denoted as the "main" version, where $t = 0.6$. It can be seen that the models with different values of $t$ achieve comparable results, outperforming the StyleGAN2 baseline. On FFHQ-5k ($256 \times 256$), $t = 0.6$ could be a more plausible choice. For a further explanation, we use $t = 0.6$ as the default value since it works well in most cases. In practice, the value of $t$ could be further adjusted to achieve even better results. Empirically, a smaller $t$ can be chosen when one has fewer data. This means the deception strength $p$ can be adjusted to increase more rapidly since the discriminator is more prone to overfitting when the data amount is fewer.

## 5 Discussion

We have shown the effectiveness of the proposed adaptive pseudo augmentation (APA) for state-of-the-art GAN training with limited data empirically. With negligible computational cost, APA achieves comparable or even better performance than other types of solutions on various datasets. APA is also complementary to existing methods based on standard data augmentations.

**Limitations.** Despite promising results, the quality of synthesized images by APA on the datasets with extremely limited data amount (*e.g.*, hundreds of images) can still be improved. Besides, on certain datasets such as FFHQ-5k, applying APA solely may be slightly inferior to approaches based on standard data augmentations. Since we do not apply any external augmentations, these two limitations are both due to the insufficiency of the dataset's intrinsic diversity. These limitations may be approached in the future in two ways: 1) Incorporating better standard data augmentations to APA. 2) Exploring the issue of data insufficiency from the generator aspect, *e.g.*, using a multi-modal generator [39] to enhance diversity. In addition, we only theoretically verified the convergence and rationality of APA. In future work, the theoretical analysis on the effectiveness of APA could be further explored.

**Broader impact.** On the one hand, the effectiveness of APA with negligible computational cost will benefit the practical deployment of GANs, especially in the low-data regime. APA may also extend the breadth and potential of solutions to training GANs with limited data and benefit downstream tasks, such as conditional synthesis. On the other hand, APA may also bring potential concerns on its capability to ease the higher-quality fake media synthesis using only limited data, as technology is usually a double-edged sword. However, we believe that these concerns can be resolved by developing better media forensics methods and datasets as countermeasures.

**Acknowledgments and funding disclosure.** This study is supported under the RIE2020 Industry Alignment Fund – Industry Collaboration Projects (IAF-ICP) Funding Initiative, as well as cash and in-kind contribution from the industry partner(s).

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
