# OpenReview forum: "Deceive D: Adaptive Pseudo Augmentation for GAN Training with Limited Data"
_NeurIPS.cc/2021/Conference — NeurIPS 2021 Poster_

### Official Review · Reviewer_Kz3j · 2021-07-15

**Rating:** 7
**Confidence:** 4

**Summary:**

This paper introduces a data augmentation strategy for training GANs, which proves to be especially effective in the low data regime. The essence of the idea is to replace real images by images produced by the generator, using an adaptive replacement probability in the discriminator training step. By classifying some fake images as real, discriminator overfitting is prevented, which normally leads to very poor training performance and instability when very little data is available. The proposed method (named APA) improves over or can be combined with another recent data augmentation scheme for the low data regime (such as ADA).

**Limitations And Societal Impact:**

yes, the limitations are addressed in the paper

**Main Review:**

Strengths:

-	The proposed method (APA) is simple and can be easily applied to other data modalities, besides images, in contrast to ADA which requires image-specific transformations. At the same time, it performs better most of the time and can be applied in combination with ADA.
-	The proposed approach is quite effective and helps to stabilize the training under limited training data.
-	The paper is sound and easy to read.


Weaknesses:

-  The core idea of the proposed method is adding label noise to the discriminator (switching the label of fake images to real), which is a well-known trick used for improving training stability of GANs [https://github.com/soumith/ganhacks]. One way was to flip the labels of the real images, adding pseudo-labels to the fake images, or to flip the labels of both fake and real with the fixed ratio. In the proposed approach, the labels are flipped for the fake images only. It’d be beneficial to see if the comparison can be provided between one-sided flipping techniques (adding pseudo-labels to either real or fake images) and a two-sided label flipping (adding pseudo-labels to both real or fake images). Also the paper should state clear that adding label noise is a well known technique to improve the stability of the GAN training.

-	There are also other techniques that can “cripple” discriminator, such as one-sided label smoothing [37], instance noise [39] or progressive augmentation of the discriminator [PA, Zhang et al., NeurIPS’2019]. As the proposed APA approach is quite close to those techniques, I think the comparison with them is required.

-	The clarity of  Section 3.1 (Method) can be improved: the description of the adjustment of t is a little vague. Take the sentence “If lambda signifies too much/little overfitting regarding t, the probability p will be increased/decreased by one fixed step.”. In particular, how much is too much overfitting regarding t? What exactly is the step size? It seems like the step size is chosen such that p increases to 1 after exactly 500K images or it is increased in every single step?

-	It would be beneficial to see how the APA method performs under low amount of training data, when the number samples is less than 1k images.

-	Table 3: What happens if you combine ADA and APA on the full dataset?

-	In Section 4.3, it would be helpful to see an ablation on the hyper-parameter t. How was threshold t=0.6 chosen?



**Time Spent Reviewing:**

5h

---

> ### Author Response · Authors · 2021-08-10
> **Author Response to Reviewer Kz3j**
>
> Thanks for the constructive suggestions. Below we address the concerns.
>
> It is worth mentioning that the goal of APA is different from the standard data augmentations, which is not to enlarge the real dataset but to suppress the discriminator’s confidence in distinguishing real and fake distributions (see Line 38 - Line 43). Thanks for pinpointing the advantages of APA over previous methods.
>
> **1. Connection with label flipping and other tricks for training GANs**
>
> Indeed, APA is relevant with label flipping and other strategies for training GANs. We sincerely thank you for providing valuable suggestions. However, there are several clear differences between APA and these methods. Reviewer e1RP highlighted these differences, and we further summarize them as follows.
>
> - APA and these methods are designed for different goals. Previous strategies are mainly used for stabilizing training or preventing mode collapse. However, APA is specialized for training GANs in the low-data regime.
> - As mentioned by Reviewer e1RP, APA is an effective practice and improvement of these ideas on modern GANs, whose implementations are very different from the early ones.
> - Compared to previous techniques, APA is more adaptive to fit different settings and training status. As mentioned by Reviewer e1RP, although using adaptive heuristics was also explored in the past, it had been found unpractical at the time. APA makes the adaptive control scheme possible in practice.
>
> We believe that the proposed APA could contribute to the community for its effectiveness, simplicity, and adaptability for training state-of-the-art GANs in the low-data regime. It is a good suggestion to state clear these strategies and their connections with APA. Some discussions were provided in Line 82 - Line 85. We will include more discussions in the revision.
>
> For completeness, we follow the suggestions and provide additional studies on APA and these tricks. First, the comparison between one-sided label flipping (analogous to APA) and two-sided label flipping is interesting. For fairness, we only ablate the flipping strategy and keep other parts the same. The results are reported below. Empirically, we observe that two-sided label flipping can still outperform StyleGAN2 but is inferior to one-sided flipping.
>
> - Comparison between one-sided and two-sided label flipping on FFHQ-5k ($256 \times 256$):
>
> |Method|FID$\downarrow$ (FFHQ-5k, ~7% data)|IS$\uparrow$ (FFHQ-5k, ~7% data)|
> |:---|:---:|:---:|
> |StyleGAN2|37.830|4.018|
> |Two-sided flipping|15.440|4.167|
> |One-sided flipping (Ours)|**13.249**|**4.487**|
>
> Besides, the comparison with other techniques that can "cripple" the discriminator could be beneficial. We provide representative results below. It can be seen that applying instance noise may not boost the performance of StyleGAN2 so much under limited data. One-sided label smoothing (with the real label of 0.9) achieves better results than StyleGAN2 but still has a huge performance gap with the proposed APA. This further suggests the effectiveness and usefulness of APA.
>
> - Comparison with other techniques that "cripple" the discriminator on FFHQ-5k ($256 \times 256$):
>
> |Method|FID$\downarrow$ (FFHQ-5k, ~7% data)|IS$\uparrow$ (FFHQ-5k, ~7% data)|
> |:---|:---:|:---:|
> |StyleGAN2|37.830|4.018|
> |Instance noise|40.981|4.231|
> |One-sided label smoothing|33.978|4.029|
> |APA (Ours)|**13.249**|**4.487**|
>
> We will provide more results and analysis in the revision.
>
> **2. The description of the adjustment of $p$ can be clearer**
>
> Our code will be released to ensure reproducibility. First, the adjustment is applied to $p$ but not $t$. "$\lambda$ signifies too much/little overfitting regarding $t$" means the current value of $\lambda$ is larger/smaller than $t$ (in most cases of our experiments, $t=0.6$). Besides, the step size is chosen so that $p$ can increase from $0$ to $1$ if $p$ keeps increasing using this step size during shown 500k images. In addition, $p$ is increased in every single step. We missed one small detail that we adjust $p$ once every four iterations (results are not sensitive to this parameter) in practice, which will be included in the revision. For instance, if the batch size is $64$, the exact step size every four iterations is: $1 \div 500000 \times 64 \times 4 = 0.000512$. Thanks for the suggestion. We will make our description clearer.
>
> **3. The performance of APA under the lower amount of data, *i.e.,* fewer than 1k**
>
> We agree that showing the performance of APA under the lower amount of data could be interesting. Thus, below we provide more results on FFHQ-500 (a subset of 500 images, ~0.7% of full data) and the transfer learning results on MetFaces-500 \[17\] (a subset of 500 images, ~37% of full data) from the pre-trained model on FFHQ-70k. The resolution is $256 \times 256$. Under fewer data, APA can still boost StyleGAN2 performance by a large margin. However, the quality itself of synthesized images by APA can still be improved, consistent with our discussion on limitations in Line 263 - Line 264. Still, including these results in the revision is beneficial.
>
> - Results under the data amount fewer than 1k:
>
> |Method|FID$\downarrow$ (FFHQ-500, ~0.7%)|IS$\uparrow$ (FFHQ-500, ~0.7%)|FID$\downarrow$ (MetFaces-500, ~37%)|IS$\uparrow$ (MetFaces-500, ~37%)|
> |:---|:---:|:---:|:---:|:---:|
> |StyleGAN2|119.815|2.446|54.691|3.218|
> |APA (Ours)|**50.989**|**4.099**|**29.508**|**3.986**|
>
> **4. The performance of ADA+APA on the full dataset**
>
> If we combine ADA and APA on the full dataset, the model obtains the FID score of $3.811$ and the IS score of $5.321$, which are better than the baseline but slightly inferior to applying the proposed APA solely. The degraded performance is mainly affected by ADA, which we empirically found might slightly harm the performance when the training data is sufficient. Overall, these methods are not dedicated to improving performance under sufficient data. Nevertheless, as a beneficial side effect, the proposed APA may have this potential.
>
> **5. Ablation study on the hyperparameter $t$**
>
> We provide the additional ablation study on the hyperparameter $t$ below. The "main" denotes the main version used in the experiments of our main paper. It can be seen that the models with different values of $t$ achieve comparable results, outperforming the StyleGAN2 baseline. On FFHQ-5k ($256 \times 256$), $t=0.6$ could be a more plausible choice.
>
> - Ablation study on the hyperparameter $t$ on FFHQ-5k ($256 \times 256$):
>
> |Method|FID$\downarrow$ (FFHQ-5k, ~7% data)|IS$\uparrow$ (FFHQ-5k, ~7% data)|
> |:---|:---:|:---:|
> |StyleGAN2|37.830|4.018|
> |$t=0.4$|13.687|4.418|
> |$t=0.6$ (main)|**13.249**|**4.487**|
> |$t=0.8$|13.984|4.395|
>
> For a further explanation, we simply use $t=0.6$ as the default value since it works well in most cases. In practice, the value of $t$ could be further adjusted to achieve even better performance. Empirically, a smaller $t$ can be chosen when one has fewer data. This means the deception strength $p$ can be adjusted to increase more rapidly since the discriminator is more prone to overfitting when the data amount is fewer. We will include more results and discussions in the revision.

---

> > ### Comment · Reviewer_Kz3j · 2021-08-24
> > **Response to rebuttal**
> >
> > Thanks a lot for your detailed feedback, provided extra experiments and all clarifications! The rebuttal has mostly addressed my concerns.
> >
> > I think it should be clearly stated in the final version of the paper that adding label noise is a well-known GAN technique which helps to prevent overfitting of the discriminator, and that there are also other techniques which serve the same purpose such as label smoothing, adding instance noise or progressive augmentation of the discriminator.
> >
> > I don't fully agree with the provided argumentation that the APA and the above mentioned methods are designed for different goals. The main purpose of these techniques is to prevent the discriminator being overconfident, which leads to training instabilities. With more evolved GAN architectures, this is more often observed while training with limited data. Therefore, I agree with the comment of Reviewer GodP that "GANs in the low-data regime had similar symptoms and thus the goal should be considered the same". Including the comparison with these techniques would make the paper stronger.
> >
> > Thus, I think the prior related techniques (label smoothing, instance noise, PA of the discriminator...) as well as the proximity between the objectives of training in low-data regime and reducing GAN instability should be clearly stated and discussed in the revised paper. Assuming that all the stated changes are made to the final version, I would be happy to accept this work.

---

> > > ### Author Response · Authors · 2021-08-25
> > > **Author Response to Additional Comments by Reviewer Kz3j**
> > >
> > > We sincerely thank Reviewer Kz3j for raising the score and providing additional valuable suggestions. Indeed, APA is closely related to adding label noise and some other techniques for improving the stability of GAN training. We agree that clearly stating these techniques and the proximity between the objectives of training in the low-data regime and reducing GAN instability can further strengthen our paper. We provided some additional comments in our further discussion with Reviewer GodP. We shall follow the suggestion by Reviewer Kz3j to include more detailed discussions and comparisons on APA and these techniques in our revised paper.

---

### Official Review · Reviewer_zQEC · 2021-07-16

**Rating:** 7
**Confidence:** 4

**Summary:**

This paper proposes a new method for GAN training with limited data, named adaptive pseudo augmentation (APA). The key idea originates from the empirical observation that, when trained with limited data, the GAN discriminator tends to overfit easily, leading to poor generation performance. Therefore, the authors propose to adaptively add generated samples to the training data, so as to counteract the overfitting during training. Accordingly, they present the APA that is developed based on the output of the discriminator. Empirical experiments demonstrate the effectiveness of the proposed APA.

**Limitations And Societal Impact:**

Yes.

**Main Review:**

Originality. The presented APA is believed new and sound. More importantly, APA is quite easy to implement in practice. Related work is adequately cited and discussed. However, in most of the experiments, APA is only compared with StyleGAN2 that is not tailored for a generation with limited data. The ADA results should be reported, for example in Table 2, to highlight the advantages of the presented APA.

Quality. The presented APA is sound and seems to be a simple but powerful technique. The claims are well supported empirically. However, theoretically, it would be better if the authors discuss why applying APA leads to better performance than the vanilla GAN? After all, the optimum is the same.

Clarity. The paper is clearly written and quite easy to follow.

Significance. The presented APA is important for GAN training with limited data; it's likely to be reused by other researchers.

Other comments:

In the experiments, why did you use Lanczos filter to do the preprocessing? Please elaborate on it.

*********************************
After rebuttal
*********************************
Thanks for the detailed responses. Please address the concerns in the comments and revise the paper accordingly.


**Time Spent Reviewing:**

2

---

> ### Author Response · Authors · 2021-08-10
> **Author Response to Reviewer zQEC**
>
> Thanks for the insightful suggestions. We answer the questions as follows.
>
> **1. More comparison with ADA**
>
> In Table 3, we provided representative comparison results of the proposed APA with other state-of-the-art solutions (*e.g.,* ADA) tailored for GAN training with limited data. We used datasets (*i.e.,* AFHQ-Cat, FFHQ-5k, and FFHQ-70k) varying in types and data amounts. We also compared the computational cost of APA against ADA in Section 4.2.
>
> Thanks for the suggestion. We agree that adding more comparison with ADA could strengthen the paper further. Below we report the additional comparison results on FFHQ-7k and FFHQ-1k (see Table 3 for 5k and 70k). To make the comparison settings more comprehensive, we also provide the transfer learning results on MetFaces \[17\] from the pre-trained model on FFHQ-70k. We have also tried a fewer data amount (*i.e.,* 500 images). The resolution is $256 \times 256$. Combined with Table 3, the proposed APA achieves comparable or even better results than ADA while with less computational cost. Both methods outperform the StyleGAN2 baseline under limited data. Although applying APA solely may be inferior to ADA on FFHQ-1k (in line with our discussion in Line 265 - Line 270), it is worth mentioning that APA is also complementary to ADA, which is very important to boost the performance further.
>
> - Additional comparison results with ADA (the best value in bold, and the second best in italic):
>
> |Method|FID$\downarrow$ (FFHQ-7k, 10%)|IS$\uparrow$ (FFHQ-7k, 10%)|FID$\downarrow$ (FFHQ-1k, ~1.4%)|IS$\uparrow$ (FFHQ-1k, ~1.4%)|FID$\downarrow$ (MetFaces-1336, full)|IS$\uparrow$ (MetFaces-1336, full)|FID$\downarrow$ (MetFaces-500, ~37%)|IS$\uparrow$ (MetFaces-500, ~37%)|
> |:---|:---:|:---:|:---:|:---:|:---:|:---:|:---:|:---:|
> |StyleGAN2|27.738|4.264|86.407|2.806|30.988|3.719|54.691|3.218|
> |ADA|*10.275*|4.813|*22.590*|*4.239*|*20.834*|4.005|30.368|3.974|
> |APA (Ours)|10.800|*4.860*|45.192|4.130|21.050|*4.103*|*29.508*|*3.986*|
> |ADA+APA (Ours)|**7.333**|**4.994**|**18.892**|**4.316**|**18.865**|**4.207**|**28.408**|**4.044**|
>
> We will provide more comparison with ADA in the revision.
>
> **2. Discussion on the limitation of theoretical analysis**
>
> Our theoretical analysis just verified the convergence and rationality of APA, and we empirically found the effectiveness of APA for training GANs with limited data.
> We do agree that in future work, we could provide the theoretical analysis on APA's effectiveness as an improvement.
> Thanks for the advice. As mentioned by Reviewer e1RP, it is a good idea to discuss the limitation of theoretical analysis in the current version. We will revise it accordingly.
>
> **3. The reason for using Lanczos filter for data preprocessing**
>
> Lanczos filter is a high-quality filter that we used to better preserve the quality of real images when performing image resizing. Although applying the Lanczos filter for data preprocessing may be slightly slower than other filters such as the Bilinear or Bicubic filters, the image quality is better. The Lanczos filter has been used in the official PyTorch implementation of ADA.

---

> ### Author Response · Authors · 2021-08-25
> **Author Response to Additional Comments by Reviewer zQEC**
>
> Thanks a lot for raising the score. We shall revise the paper based on the comments.

---

### Official Review · Reviewer_GodP · 2021-07-16

**Rating:** 8
**Confidence:** 5

**Summary:**

This paper tries to solve discriminator overfitting problem.

The authors propose adaptive pseudo augmentation (APA).
* APA employs the generator itself to augment the real data distribution with fake images.
  * i.e., Fake images are presented as "real" instances to the discriminator.
* Adaptiveness comes from measuring overfittingness of the discriminator.
* APA has similar theoretical properties to the original GAN, but with $(1-\alpha)p_\text{data} + \alpha p_g = p_g$.

APA improves systnesis quality in the limited-data regime.
* on FFHQ, AFHQ-Cat, CUB, Danbooru
The authors provide a theoretical analysis of APA, similar to the original GAN.


**Limitations And Societal Impact:**

Yes to both.

**Main Review:**

Originality.
* (+) This paper proposes a novel way to augment the real distribution by adding fake samples.
  * Existing methods transforms images to augment the real and/or fake distributions.

Quality.
* (+) APA has sound theoretical groundings.
* (+) APA improves StyleGAN2 in all cases (number of real samples).
* (+) APA is roughly robust to choice of hyperparameters.
* (-) Analyses are provided only on FFHQ 7k. Showing the similar trend on 1k and 5k would be better.
* (-) Benchmark datasets are not as diverse as ADA.
* (-) ADA is compared only on AFHQ-Cat, FFHQ-5k on FFHQ-70k.

Clarity.
* (+) Everything is clear to understand.

Significance.
* (+) APA is generally effective on different settings, even with existing standard augmentation methods.
* (+) APA brings negligible computational cost.
* (+) APA is simple.

**Time Spent Reviewing:**

1.5

---

> ### Author Response · Authors · 2021-08-10
> **Author Response to Reviewer GodP**
>
> Thanks for the valuable comments. We address the issues as follows.
>
> **1. More overfitting and convergence analysis on FFHQ-5k and FFHQ-1k would be helpful**
>
> We took FFHQ-7k as an example for the overfitting and convergence analysis since it was representative to demonstrate the effectiveness of APA in curbing the overfitting of the discriminator and improving the training convergence under limited data. The trend on FFHQ-5k and FFHQ-1k is similar to FFHQ-7k. Thanks for the great suggestion. We do agree that showing more analysis on 5k and 1k can be helpful. Due to the rebuttal policy, we will provide the figures in the revision.
>
> **2. Benchmark datasets are not as diverse as ADA**
>
> In our experiments, we have already evaluated APA on various representative datasets, covering human faces (FFHQ; aligned; natural images), animal faces (AFHQ-Cat; aligned; natural images), birds (CUB; unaligned; natural images), and anime portraits (Danbooru2019 Portraits; unaligned; unnatural images). Besides, we exploited some of their artificially limited subsets with different data amounts under diverse settings. In addition, we tried different resolutions: $256 \times 256$ and $1024 \times 1024$ (provided in the supplementary material).
>
> Nevertheless, we agree that enriching our benchmark datasets can strengthen the paper further. We have been improving the benchmark since submission. We report the BigGAN class-conditional image synthesis results on CIFAR-10 and CIFAR-100 below. It can be observed that APA outperforms BigGAN under limited training data in all cases.
>
> - CIFAR-10 ($32 \times 32$):
>
> |Method|FID$\downarrow$ (full data)|IS$\uparrow$ (full data)|FID$\downarrow$ (20% data)|IS$\uparrow$ (20% data)|FID$\downarrow$ (10% data)|IS$\uparrow$ (10% data)|
> |:---|:---:|:---:|:---:|:---:|:---:|:---:|
> |BigGAN|9.531|9.078|22.024|8.343|44.061|7.589|
> |APA (Ours)|**8.283**|**9.362**|**15.316**|**8.822**|**25.987**|**8.410**|
>
> - CIFAR-100 ($32 \times 32$):
>
> |Method|FID$\downarrow$ (full data)|IS$\uparrow$ (full data)|FID$\downarrow$ (20% data)|IS$\uparrow$ (20% data)|FID$\downarrow$ (10% data)|IS$\uparrow$ (10% data)|
> |:---|:---:|:---:|:---:|:---:|:---:|:---:|
> |BigGAN|13.281|10.525|35.590|8.706|64.828|6.635|
> |APA (Ours)|**11.429**|**11.243**|**23.506**|**9.811**|**45.794**|**8.114**|
>
> Furthermore, below we show the transfer learning results on MetFaces \[17\] from the pre-trained model on FFHQ-70k. All the metrics are boosted by APA, further verifying its effectiveness.
>
> - MetFaces ($256 \times 256$):
>
> |Method|FID$\downarrow$ (1336 img, full data)|IS$\uparrow$ (1336 img, full data)|FID$\downarrow$ (500 img, ~37% data)|IS$\uparrow$ (500 img, ~37% data)|
> |:---|:---:|:---:|:---:|:---:|
> |StyleGAN2|30.988|3.719|54.691|3.218|
> |APA (Ours)|**21.050**|**4.103**|**29.508**|**3.986**|
>
> We will include more results in the revision.
>
> **3. More comparison with ADA**
>
> In Table 3, we provided representative comparison results of the proposed APA with other state-of-the-art solutions (*e.g.,* ADA) tailored for GAN training with limited data. We used datasets (*i.e.,* AFHQ-Cat, FFHQ-5k, and FFHQ-70k) varying in types and data amounts. We also compared the computational cost of APA against ADA in Section 4.2.
>
> Thanks for the suggestion. We agree that adding more comparison with ADA could strengthen the paper further. Below we report the additional comparison results on FFHQ-7k and FFHQ-1k (see Table 3 for 5k and 70k). To make the comparison settings more comprehensive, we also provide the transfer learning results on MetFaces \[17\] from the pre-trained model on FFHQ-70k. We have also tried a fewer data amount (*i.e.,* 500 images). The resolution is $256 \times 256$. Combined with Table 3, the proposed APA achieves comparable or even better results than ADA while with less computational cost. Both methods outperform the StyleGAN2 baseline under limited data. Although applying APA solely may be inferior to ADA on FFHQ-1k (in line with our discussion in Line 265 - Line 270), it is worth mentioning that APA is also complementary to ADA, which is very important to boost the performance further.
>
> - Additional comparison results with ADA (the best value in bold, and the second best in italic):
>
> |Method|FID$\downarrow$ (FFHQ-7k, 10%)|IS$\uparrow$ (FFHQ-7k, 10%)|FID$\downarrow$ (FFHQ-1k, ~1.4%)|IS$\uparrow$ (FFHQ-1k, ~1.4%)|FID$\downarrow$ (MetFaces-1336, full)|IS$\uparrow$ (MetFaces-1336, full)|FID$\downarrow$ (MetFaces-500, ~37%)|IS$\uparrow$ (MetFaces-500, ~37%)|
> |:---|:---:|:---:|:---:|:---:|:---:|:---:|:---:|:---:|
> |StyleGAN2|27.738|4.264|86.407|2.806|30.988|3.719|54.691|3.218|
> |ADA|*10.275*|4.813|*22.590*|*4.239*|*20.834*|4.005|30.368|3.974|
> |APA (Ours)|10.800|*4.860*|45.192|4.130|21.050|*4.103*|*29.508*|*3.986*|
> |ADA+APA (Ours)|**7.333**|**4.994**|**18.892**|**4.316**|**18.865**|**4.207**|**28.408**|**4.044**|
>
> We will provide more comparison with ADA in the revision.

---

> > ### Comment · Reviewer_GodP · 2021-08-11
> > **Thank you for taking my suggestions seriously. But I am keeping my score.**
> >
> > The author response provides satisfactory facts and promises.
> >
> > However, the other reviews reminds me of the label smoothing techniques in the past. I think the discussion should be more detailed.
> > > APA and these methods are designed for different goals. Previous strategies are mainly used for stabilizing training or preventing mode collapse. However, APA is specialized for training GANs in the low-data regime.
> >
> > Above discussion is pointless because GANs in the low-data regime had similar symptoms and thus the goal should be considered the same except the data setting.
> >
> > Making this discussion clear and promising its inclusion would raise my score.

---

> > > ### Author Response · Authors · 2021-08-11
> > > **Further Discussion on APA and Previous Strategies for GAN Training**
> > >
> > > We sincerely thank Reviewer GodP for the positive feedback and additional insightful suggestion. Indeed, APA is closely related to label smoothing and some other techniques for training GANs. We do agree that providing further discussions on APA and these techniques could strengthen our paper further.
> > >
> > > We agree that training GANs in the low-data regime may exhibit similar behaviors as previously observed in early GANs with sufficient data. However, there are several differences we wish to highlight.
> > >
> > > - As mentioned in the additional comments of Reviewer GodP, the data setting is different. This difference is quite important since it is the core problem we would like to address. Even the performance of state-of-the-art StyleGAN2 deteriorates when trained with a limited amount of data, although it has exploited many advanced techniques for stabilizing training or preventing mode collapse, such as R1 regularization [1]. The main challenge that lies within the low-data regime is the overfitting of the discriminator. Although this issue might also appear on early GANs, it becomes more severe when data is limited. At least, previous methods did not consider the low-data regime carefully, and a systematic study about this issue was missing.
> > >
> > > - To further support our discussion above, please refer to our response to Reviewer Kz3j for additional comparative studies on APA and conventional techniques. Empirically, we showed that previous methods cannot boost the performance of StyleGAN2 much under limited data. Some of them showed a huge performance gap in comparison to the proposed APA. This indicates that previous relevant strategies cannot handle the low-data regime well, further suggesting the effectiveness and usefulness of APA.
> > >
> > > - Last but not least, APA and these techniques themselves are not exactly the same. First, APA is an effective practice and improvement of these ideas on modern GANs, whose implementations are very different from the early ones. Besides, compared to previous techniques, APA is more adaptive to fit different settings and the overfitting status in training. As mentioned by Reviewer e1RP, although using adaptive heuristics was also explored in the past, it had been found unpractical at the time. APA makes the adaptive control scheme possible in practice.
> > >
> > > We believe that the proposed APA could contribute to the community for its effectiveness, simplicity, and adaptability for training state-of-the-art GANs in the low-data regime. It is a good suggestion to state clear these strategies and their connections with APA. Some discussions were provided in Line 82 - Line 85. We shall follow the suggestion to include more detailed discussions and studies in the revision.
> > >
> > > ---
> > > [1] Lars Mescheder, Andreas Geiger, et al. Which Training Methods for GANs do actually Converge? In ICML, 2018.

---

> > > > ### Comment · Reviewer_GodP · 2021-08-12
> > > > **Mostly agreed.**
> > > >
> > > > I agree that the discussions in the response will make the paper stronger.
> > > >
> > > > It is great to see that the additional comparisons support the superiority of APA over previous techniques, but one more curiosity arises. Evaluations on the generated images are not enough evidence for preventing overfitting. Do you observe similar rank in the effectiveness to the overfitting problem? Adding comparison between APA and previous techniques on the discriminator outputs (Figure 6 in the main paper) to the appendix and mentioning the plot in the discussion would be perfect. If similar rank is not observed, adding some conjecture would be beneficial to the community.
> > > >
> > > > Thanks for your faithful response.

---

> > > > > ### Author Response · Authors · 2021-08-12
> > > > > **More Overfitting Analysis on APA and Previous Techniques for GAN Training**
> > > > >
> > > > > We agree that adding more overfitting analysis between APA and previous techniques on the discriminator outputs can be beneficial. The answer to the question is: Yes, we observed a similar rank of these methods in the effectiveness to counteract overfitting. Specifically, StyleGAN2 experiences diverged predictions most rapidly, and APA obtains the most effective restriction on the divergence of discriminator outputs. Besides, previous techniques are in between StyleGAN2 and APA, ranking in line with their generation performance (applying instance noise is very close to StyleGAN2). This further suggests: 1) the importance of addressing the discriminator overfitting for training GANs with limited data; 2) the effectiveness of APA in mitigating the discriminator overfitting, outperforming previous strategies.
> > > > >
> > > > > Thanks for the suggestion. We shall provide the plot of this analysis in the appendix and mention the plot in our revised discussion.

---

### Official Review · Reviewer_e1RP · 2021-07-16

**Rating:** 7
**Confidence:** 4

**Summary:**

The submission tackles the problem of training GANs with limited data.
It proposes to do so by adaptively regularizing GAN discriminators by replacing some real data by generated samples when the discriminator overfits, accoding to an overfitting heuristic.

A study of the proposed objectives show that the method keep the classical GAN optima while alleviating overfitting of the discriminator.

The method is tested on various datasets with limited data, and compared, using a StyleGAN-2 backbone, against other state-of-the-art methods for training with limited data.

**Limitations And Societal Impact:**

One limitation that is not discussed is about the possible inadequacy of a theoretical analysis at optimality to draw conclusions about the effectiveness of the regularisation during training. This approach, however, is fairly common in works about GANs.

Otherwise, limitations and societal impacts are adequately discussed.

--------------------------------------------------------
Post-rebuttal:
I'd like to thank the authors for providing the additional clarification and additional results in their rebuttal.
After reading the different reviews and responses, I keep my positive score. The method is simple and effective, backed with strong empirical evidence.
I also concur with reviewer Kz3j's suggestion to clearly state and discuss the relation of the proposed method with related techniques (label smoothing, instance noise...), as well as the proximity between the objectives of training in low-data regime and reducing GAN instability.
In that regard, the results provided in the response to reviewer Kz3j and the discussions with reviewer GodP are already very interesting additions.


**Main Review:**


Originality:
It might be worth mentioning that feeding the discriminator with generated data with the wrong label (label flipping) was a fairly well-known trick in 2016, discussed for example at the NeurIPS 2016 GAN Workshop [1, 2]. Connexions [3] had been drawn with label smoothing [4] and instance noise [5]. Using adaptive heuristics is also discussed in the same workshop presentation, and had been found unpractical at the time.
Of course, the proposed work is no less original, as modern GAN training dynamics have little to do with early implementations, they did not consider training in the low-data regime, and in any case, a detailed, peer-reviewed, report on label-flipping was missing. Still, discussing those connexions might be interesting.

Quality:
The submission is technically strong. The proposed heuristic seems very ad hoc, but the method is demonstrated to be effective in a variety of settings, including when combined with other methods. Experiments are thorough, with multiple datasets and numbers of samples.
It is however tested only with the StyleGAN-2. As it is perhaps the most popular GAN variant, the study is still very interesting and relevant but could be even stronger if tried with BigGAN as well.

Clarity:
The submission is clear and easy to understand.

Significance:
The tackled problem is important (training with limited data) and the proposed method is a simple and very effective method that is likely to be of use to the community.

----------
[1] How to train a GAN?
Soumith Chintala. NIPS 2016 Workshop on Adversarial Training
https://sites.google.com/site/nips2016adversarial/

[2] soumith. https://github.com/soumith/ganhacks#6-use-soft-and-noisy-labels

[3] Instance Noise: A trick for stabilising GAN training
Ferenc Huszár. inFERENCe.
https://www.inference.vc/instance-noise-a-trick-for-stabilising-gan-training/

[4] Improved Techniques for Training GANs.
Tim Salimans, Ian J. Goodfellow, Wojciech Zaremba, Vicki Cheung, Alec Radford, Xi Chen. NeurIPS 2016.

[5] Amortised MAP Inference for Image Super-resolution.
 Casper Kaae Sønderby, Jose Caballero, Lucas Theis, Wenzhe Shi, Ferenc Huszár. ICLR 2017.

**Time Spent Reviewing:**

2

---

> ### Author Response · Authors · 2021-08-10
> **Author Response to Reviewer e1RP**
>
> Thanks for the insightful comments. Below we address the issues.
>
> **1. Connection with label flipping and other tricks for training GANs**
>
> Indeed, APA is relevant with label flipping and other strategies for training GANs. We sincerely thank you for providing useful references and pinpointing the differences between APA and these approaches. We further summarize their differences as follows.
>
> - APA and these methods are designed for different goals. Previous strategies are mainly used for stabilizing training or preventing mode collapse. However, APA is specialized for training GANs in the low-data regime.
> - APA is an effective practice and improvement of these ideas on modern GANs, whose implementations are very different from the early ones.
> - Compared to previous strategies, APA is more adaptive to fit different settings and training status. As mentioned by the reviewer, although using adaptive heuristics was also explored in the past, it had been found unpractical at the time. APA makes the adaptive control scheme possible in practice.
>
> Please refer to our response to Reviewer Kz3j for additional studies on APA and these tricks. We agree that discussing those connections can be interesting. Some discussions were provided in Line 82 - Line 85. We will include more discussions in the revision.
>
> **2. More results on BigGAN**
>
> Due to the time and resource constraint to conduct large-scale GAN training, we only provided results on the most popular state-of-the-art GAN variant, StyleGAN2. However, we have been devoting our effort to enriching our benchmark further since submission. We report the BigGAN class-conditional image synthesis results on CIFAR-10 and CIFAR-100 below. It can be observed that APA outperforms BigGAN under limited training data in all cases.
>
> - CIFAR-10 ($32 \times 32$):
>
> |Method|FID$\downarrow$ (full data)|IS$\uparrow$ (full data)|FID$\downarrow$ (20% data)|IS$\uparrow$ (20% data)|FID$\downarrow$ (10% data)|IS$\uparrow$ (10% data)|
> |:---|:---:|:---:|:---:|:---:|:---:|:---:|
> |BigGAN|9.531|9.078|22.024|8.343|44.061|7.589|
> |APA (Ours)|**8.283**|**9.362**|**15.316**|**8.822**|**25.987**|**8.410**|
>
> - CIFAR-100 ($32 \times 32$):
>
> |Method|FID$\downarrow$ (full data)|IS$\uparrow$ (full data)|FID$\downarrow$ (20% data)|IS$\uparrow$ (20% data)|FID$\downarrow$ (10% data)|IS$\uparrow$ (10% data)|
> |:---|:---:|:---:|:---:|:---:|:---:|:---:|
> |BigGAN|13.281|10.525|35.590|8.706|64.828|6.635|
> |APA (Ours)|**11.429**|**11.243**|**23.506**|**9.811**|**45.794**|**8.114**|
>
> More results will be added to the revision.
>
> **3. Discussion on the limitation of theoretical analysis**
>
> Our theoretical analysis just verified the convergence and rationality of APA, and we empirically found the effectiveness of APA for training GANs with limited data.
> We do agree that in future work, we could provide the theoretical analysis on APA's effectiveness as an improvement.
> It is a good suggestion to discuss the limitation of theoretical analysis in the current version. We will revise it accordingly.

---

### Decision · Program_Chairs · 2021-09-27

**Decision:**

Accept (Poster)

**Comment:**

This paper proposes a very simple method of adaptively feeding generated instances into real data to prevent the overfitting of GAN Discriminator in the situation where the training data is limited. It is somewhat incremental in terms of novelty, considering existing label-noise techniques, and the algorithm in deciding the feeding probability is also somewhat heuristic. The theory also shows a trivial extension from original GAN paper. Nevertheless, all reviewers, including this AC, unanimously supported the acceptance of this paper because of its simplicity and strong empirical success. In the final version, please be sure to add comparison/discussion with the related technique mentioned by reviewers.